# Flexible Hybrid Electrodes for Continuous Measurement of the Local Temperature in Long-Term Wounds

**DOI:** 10.3390/s21082741

**Published:** 2021-04-13

**Authors:** Ana María Rodes-Carbonell, Juan Torregrosa-Valls, Antonio Guill Ibáñez, Alvaro Tormos Ferrando, María Aránzazu Juan Blanco, Antonio Cebrián Ferriols

**Affiliations:** 1Instituto Tecnológico del Textil (AITEX), 03801 Alcoy, Spain; 2Faculty of Medicine and Health Sciencies, Universidad Católica de Valencia San Vicente Martir, 46001 Valencia, Spain; juan.torregrosa@ucv.es (J.T.-V.); ma.juan@ucv.es (M.A.J.B.); 3Electronic Engineering Department, Universitat Politècnica de València, 46022 Valencia, Spain; aguill@eln.upv.es (A.G.I.); atormos@eln.upv.es (A.T.F.); acebrian@eln.upv.es (A.C.F.)

**Keywords:** lower extremity wound, wound temperature, infection, inflammation, wound continuous monitoring, screen printing, printed sensors, flexible hybrid electronics

## Abstract

Long-term wounds need a continuous assessment of different biophysical parameters for their treatment, and there is a lack of affordable biocompatible devices capable of obtaining that uninterrupted flow of data. A portable prototype that allows caregivers to know the local temperature behavior of a long-term wound over time and compare it with different reference zones has been developed. Alternative flexible substrates, screen-printing techniques, polymeric inks, and an embedded system have been tested to achieve potential indicators of the status and evolution of chronic wounds. The final system is formed by temperature sensors attached to a flexible and stretchable medical-grade substrate, where silver conductive tracks have been printed as interconnections with the data-acquisition unit. In addition, a specific datalogger has been developed for this system. The whole set will enable health personnel to acquire the temperature of the wound and its surroundings in order to make decisions regarding the state and evolution of the wound.

## 1. Introduction

Long-term wounds represent one of the most important causes of morbidity and mortality for patients with chronic diseases such as diabetes or venous insufficiency. Likewise, those caused by physical factors like pressure, friction, or shearing are characteristic of bedridden patients [1]. Currently, these injuries are a challenge for professionals who apply advanced wound care, and represent a high cost to both human and economic resources [2].

Long-term wound healing requires cellular activities that need an optimal specific environment to take place in the right form and time. Wound healing strongly depends on different biophysical parameters such as temperature or pH. The proper evolution of these parameters enables the transition through the different healing phases (hemostasis, inflammation, proliferation, and remodeling/epithelialization), thereby promoting local homeostasis. On the other hand, their alteration may result in a delay of the wound healing [3], or even in an infection by indigenous or external microorganisms. The management, treatment, and monitoring of these wounds is mainly based on clinical expertise and qualitative methods with a high subjectivity level. The evolution of these biophysical parameters is not estimated quantitatively. This is because the measuring devices only allow sporadic quantification, as they require complex calibrations with a high cost. Additionally, these measurements may be biased by a difficult anatomical morphology, the environment where they are performed, the technique, and the dimensions of the probes used.

There is currently great interest in measuring these wound characteristics, since they involve potential indicators and predictors of the evolution healing [4]. The wound anatomy needs to be locally delineated. Each anatomical region will have a meaning in the subsequent interpretation of the results obtained from the measurement of the characteristics of the wound. Wounds International divides the wound into two anatomical regions of great interest: the wound bed and the periwound skin. The wound bed is the region where there is a solution of continuity and where the tissues below the epidermis are exposed, be they subcutaneous tissue, fasciae, muscles, tendons, or bone. According to Dowsett and Hallern, the perilesional skin is considered as the distance between the wound edge and the 4 centimeters of skin that surrounds the wound-bed perimeter. This region faces a series of challenges during the wound-care process due to its influence on the healing prognosis, and therefore acquires great relevance in the objective assessment of the wound. Continuous assessment of the zone around the wound through indicators can help to make objective the abstract phenomenon of healing. Quantifying the state of the wound bed, its edge, and the state of the perilesional skin will allow caregivers to dynamically assist treatment plans and to compare results, reducing the healing time. The determination of these results is the key to the research in healthcare. All the changes achieved in those indicators can be considered the results from the interventions over the wounds.

New organic substrates such as polyurethane (PU), polydimethylsiloxane (PDMS), polyethylene terephthalate (PET), indium and tin oxide (ITO), or polyaniline (PANi) allow circuit printing using techniques such as screen printing. This enables the development of continuous wound sensing and monitoring systems that require certain characteristics. A series of conductive tracks based on high-conductivity electronic inks are printed on these substrates by screen printing. These tracks serve as an interface between the sensor and the acquisition system. Following the same manufacturing technique, Milne in 2014 [5] and Rahimi in 2015 [6] proposed a pH measurement system printed on an ion-selective membrane, using a reference electrode composed of Ag/AgCl printed in the same way. Later, they developed a humidity sensor based on the same system of polymeric silver inks on PU [7]. This manufacturing technique used by the previous authors was characterized by Suikkola in 2016 [8]. On the other hand, authors such as Matzeu and Salvo [9,10] developed sensory-resistive sensors for continuous temperature measurement using materials such as polyimide (Kapton) and integrated graphene nanotubes. All these sensors met the requirements for reliability and flexibility, although their reuse and biocompatibility are not entirely known due to the use of components not tested in humans.

Although these technologies have points of similarity, the choice of different textiles and films provide differentiating characteristics between them. Flexible substrates like PU and PDMS can be reusable and sterilizable, as they are not associated with single-use wound dressings. They have elasticity and are adaptable to the body morphology. Furthermore, their biocompatibility has been demonstrated [11,12].

Among the measurable parameters that can be highlighted are pH and temperature, which Kassanos achieved to measure using electrochemical deposition [13,14]. There also are more complex ones, such as transepidermal water loss (TEWL). Of these, temperature is a simple parameter to interpret that can provide interesting information regarding the state of chronic wounds (CWs). The temperature in the wound bed should be between 31 and 35 °C, while in the perilesional skin it should be between 31 and 34 °C [15]. Temperatures around 33 °C are necessary for optimal cellular processes [16].

Armstrong et al. suggested that changes in the temperature gradients in different locations of the wound, compared to reference areas, are indicators of the presence of inflammation, infection and, in the absence of these, the evolution toward healing [3]. Feirheller et al. reported a connection between the increase in perilesional skin temperature regarding the bed temperature, and the presence of infection. They recommended the inclusion of the measurement of these gradients in the routine assessment of CWs [17].

Mufti [18] compared the most popular techniques, such as thermography and infrared thermometry, with four-point measurement protocols. Despite being reliable, both techniques only allowed discontinuous measurements and were influenced by the environment. In contrast, although contact thermometry is less used, it lacks some of the main limitations presented and allows continuous measurements of this parameter with an adequate system [19,20]. Objectification of the wound’s assessment becomes necessary to guide the treatments. Long-term wounds are a silent epidemic with a high social, professional, and economic burden, so techniques and devices are sought to reduce costs, shorten healing time, and improve the patient’s quality of life.

The ease of access and cost of manufacturing techniques mentioned above allow the manufacturing of devices able to assist in monitoring and control the status of this type of wound. The purpose of this research is to develop a continuous measurement system for temperature differentials. It will be based on flexible and elastic medical grade PU substrates that are able to adapt to the morphology of the wound. It will be marked by high autonomy, as well as low impact on the body and on the patient’s mobility. With it, it will be possible to detect critical changes related to inflammation and infection that aggravate and prolong a wound.

## 2. Materials and Methods

### 2.1. Sensor Design

A prototype has been developed to be used in chronic wounds in order to gather longitudinal information about the temperature evolution. The system allows caregivers to know the behavior of the local temperature of the wound over time and to compare it with different reference areas (proximal and perilesional).

The prototype consists of two main blocks, a temperature multisensor and an electronic system for measuring, storing, and managing data. The main features of each block are detailed in Table 1.

The multisensor block comprises the sensors that will facilitate, in contact with the wound and the reference areas, the acquisition of the temperatures of interest.

The electronic system is responsible for the quantification, storage, and management of the information gathered. It will be made up of a microprocessor for the acquisition of the signal generated by the sensors, a recording unit, and the necessary components for data transmission.

### 2.2. Multisensors

Figure 1 shows the sensor system, and the multisensor block is detailed. This part is composed of a biocompatible flexible substrate on which conductive tracks are screen-printed using high-conductivity inks, thus connecting the temperature sensors with the connectors of the electronic system. There are four sensor elements for measuring the temperature of the wound (Tu), the periwound zone (Tp), the healthy skin as reference (Tr), and the environmental (Ta). The sensors are PDMS-coated to avoid direct contact of the sensor with the injury or the skin. The rest of the substrate is also protected with a second layer of heat-sealed PU film, thus avoiding direct contact with the conductive inks. The substrates and the silver conductive inks are flexible and stretchable, so they can be adapted to any shape and length.

For the sensor-element selection, several requirements have been considered, which are detailed in Table 2.

The commercial sensor model used is the NTCS0805E3104SMT from Vishay (Malvern, PA, USA). It is a negative temperature coefficient (NTC) thermistor, the main characteristics of which are shown in Table 3. The sensor encapsulation has a surface of 2.5 mm^2^ and a low enough profile to avoid discomfort or injuries caused by pressure on the wound.

With these premises, the dimensions of the multisensor element are shown in Figure 2. The total length is 47 cm, determined by the length of the injury temperature sensor (Tu). The periwound sensor (Tp) measures 45 cm, the reference sensor (Tr) measures 31 cm, and the environmental temperature sensor (Ta) is 26 cm long. A unique eight-pin connector has been chosen to facilitate connection to the electronics. The end of the substrate has the required dimensions for the correct positioning of the connector (track size of 1.5 mm and 1 mm of pitch). The length of the connecting tracks is 30 mm. At each end where the sensors are located, a soldering zone has been designed for the NTC with an 0805 encapsulation.

The support where the sensors are located must be a flexible material, such as cotton textile, PU, or PDMS, which are gas-permeable and biocompatible. Cotton textiles for medical uses are widely used. The PU film has perfect characteristics for its use in medical applications: breathability, elasticity, and formability. The most important characteristic for application as a dressing is breathability and, at the same time, the ability to keep the covered area dry. PDMS is recommended for laboratory experiments and medical research that require a gas-permeable and liquid-impermeable membrane.

Three PU films and one PDMS film have been used as substrates to define which provides the best results, and later they will be validated in cotton textiles. Table 4 shows the main characteristics of the PU films used: The EU50 and EU94DS were from DelStar Technologies, Inc. (Middletown, DE, USA), the SK81SS + 06 was from Novotex Italiana SpA (Gaggiano, MI, Italy), and the film PDMS SSPM823-040-12 was from SSP, Inc. (Ballston Spa, NY, USA).

To print the conductive tracks on the substrate, which allow the electrical connection between the sensor and the connector, conductive silver inks (Table 5) have been screen-printed with two key features: elasticity and flexibility. PE873 from DuPont (Wilmington, DE, USA) and IPC-603X from Inkron (Kutojantie, Espoo, Finland), both with good compatibility with polymeric substrates, have been used. Neither manufacturer specifies their degree of biocompatibility.

### 2.3. Electronic System

Multisensor temperature measurements were carried out by an electronic system based on a microcontroller (Figure A1). The detailed circuit diagram is shown in Figure A2. It is a standalone system based on an ultralow-consumption 32-bit ARM Cortex STM32L151RB from STMicroelectronics (Geneva, Switzerland). Each of the four NTC sensors was excited by a 3 V voltage through a series resistance in a resistive divider configuration. Voltage was digitized in a 12-bit resolution ratiometric conversion (one sample per second for real-time measurements), converted to temperature, and stored in memory (one measurement per minute). Additionally, external temperature and humidity were recorded with an integrated Si7021 sensor from Silicon Laboratories Inc. (Austin, TX, USA). Temperatures were stored in an external flash memory using the serial peripheral interface (SPI) protocol.

The temperature measurement mode, the operating mode of the electronic system and its electrical characteristics, and the software used are shown in Appendix A.

### 2.4. Manufacturing

The manufacturing technology used was based on thick-film screen-printing technology. The screen-printing process consists of forcing pastes of different characteristics on a substrate through screens using scrapers. The openings in the screen define the pattern to be printed on the substrate. The final thickness of the pastes can be adjusted by varying the thickness of the screens.

When screen-printing technology is used, it is necessary to make frames with a mesh for the design. The screen for the conductive tracks was made of 230 PET 1500 90/230-48 mesh polyester material from Sefar (Thal, Switzerland). To transfer the pattern to the screen mesh, a UV Dirasol 132 film from Fujifilm (Minato, Japan) was used. The final thickness of the screen was 74 μm for the conductive designs screen. The pattern was transferred to the screen using an UV IC-5000 light source from BCB (Tarrasa, Spain).

Printing was carried out by using an EKRA E2XL screen printer from ASYS Group (Dornstadt, Germany) with a shore 75° hardness squeegee, 60° squeegee angle, 1 mm of snap-off, 3.5 bar of force, and a speed of 100 mm/s. Figure 3 shows the transferred pattern in each frame for printing multiple units at once.

After the deposition of inks, they were cured in a FED-115 air oven from BINDER (Tuttlingen, Germany) at 130 °C for 15 min using the same curing characteristics for both inks. Although some manufacturers recommend curing at higher temperatures, lower temperatures can be used with an increase of the cure time. In this case, it was done in this way to avoid the film deterioration during heating, since the typical melting point for PU films is 150 °C. The final result is shown in Figure 4.

The PU film’s second layer was heat-sealed on the first layer with a DCH-100 thermal press from Microtec (Valencia, Spain) at 130 °C for 60 s.

NTC thermistor welding on the PU has been a critical point, since the typical melting point is 150 °C, as mentioned. For this reason, a soldering study, which is detailed in Section 3.4, was carried out.

The NTC thermistors were covered with a thin layer of PDMS SYLGARD 184 from Sigma-Aldrich (St. Louis, MO, USA), deposited as a drop to avoid direct contact on the skin. On the other hand, PDMS provides consistency to the solder, hindering possible shear failures when the stretching of the base film occurs. Figure 5 shows the result of this process. Although PDMS thermal conductivity was very low (0.2 W/m∙K), and therefore there was some inherent thermal inertia to the NTC–PDMS block, it was not critical since the measurement was carried out every 60 s, a long enough period to stabilize the temperature in the NTC–PDMS block. Furthermore, the goal of the electrode was to measure temperature increases, rather than to accurately measure the temperature at the operation point.

Clincher flex connectors from Amphenol Corporation (Wallingford, CT, USA) were used to connect the electrodes with the measurement equipment, since they allow close contact between the conductive tracks and the connector by pressure. With this configuration, the final prototypes shown in Figure A6 weredeveloped

### 2.5. Measurements

Resistance measurements were made with a Fluke 8845A multimeter from Fluke Corp. (Everett, WA, USA).

3D profilometry was measured with a Profilm3D profilometer from Filmetrics Inc. (San Diego, CA, USA) with a Nikon CF IC Epi Plan × 20 objective.

For the tensile test on PU films, a 3400 single-column equipment from Instron (Norwood, MA, USA) was used to determine the behavior of materials under axial stretching loads (up to 5000 N). The sample length increased between 1% and 10% of its nominal length, and its resistance was measured with the Fluke 8845A multimeter (Figure 6 and Figure 7).

To measure the shear strength of welds, the V-275.431 PIMag voice-coil linear actuator with force sensor from Physik Instrument (PI) GmbH & Co. KG (Karlsruhe, Germany) was used.

## 3. Results and Discussion

### 3.1. Study of the Conductive Inks on Films

The surface characteristics of PDMS made it difficult to print conductive inks on this film. To improve printing, different surface treatments had to be used to improve the surface tension of PDMS, such as a treatment with 30% sulfuric acid or a corona treatment. Despite obtaining improvements, it was decided not to use this substrate since the final print was not adequate (Figure 8).

To calculate the average thickness of both inks, a resistance pattern of 15 × 3 mm was screen-printed on thermosetting PET film, resulting in an average layer thickness of 1.2 µm for DuPont PE873 and 7 µm for Inkron IPC-603X (Figure 9).

Conductive layer thickness before drying can be calculated using Equation (1):(1)$$T_{bd}=\left(T_{S}\cdot A_{s}\right)+T_{f}$$
where *T_bd_* is the thickness of the conductor before drying, *T_S_* is the thickness of the screen, *A_S_* is the open area of the screen, and *T_f_* is the thickness of the photosensitive film. The values used were from the display data sheet used for the 230 mesh polyester material conductors (PET 1500 90/230-48 from Sefar) and the UV Dirasol 132 film from Fujifilm. The value obtained for *T_bd_* was 21.75 µm (*T_S_* = 71 µm, *A_S_* = 25%, and *T_f_* = 4 µm).

*T_bd_* is reduced after drying according to the percentage of solid content and the type of ink solvent. The reduction after drying was 94.5% for the DuPont ink and 67.8% for the Inkron ink.

Then, the actual resistivity value was calculated and compared with the sheet resistivity provided by the manufacturer.

Equations (2) and (3) make the comparison possible:(2)$$R_{S}=\varrho \frac{L}{t\cdot W}\to \varrho =\frac{R_{S}\cdot t\cdot W}{L}$$
(3)$$\varrho_{Sheet}=\frac{\varrho }{25\mu m}$$
where *R_S_* is the resistance of the sample (1.1 Ω in the case of DuPont ink and 0.35 Ω in the case of Inkron ink), ρ is the resistivity, *ϱ_sheet_* is the manufacturer’s sheet resistivity, t is the layer thickness, L is the length, and W is the resistance width. The value of 25 µm is the print thickness used by manufacturers to specify the resistivity of the sheet.

Therefore, for DuPont PE873 ink, the resulting sheet resistivity was 7.5 mΩ/sq/mil (according to the manufacturer, <75 mΩ/sq/mil), and for the Inkron IPC-603X ink, the resistance of the sheet resistivity was 16 mΩ/sq/mil (according to the manufacturer, <15 mΩ/sq/mil).

### 3.2. Study of the Polyurethane Films on Fabrics

To test the elongation effect on screen-printed conductive inks printed on PU and PDMS films, a test was carried out with patterns of different widths. The test checked the pattern value according to the percentage of elongation. In the case of PDMS, the results were not conclusive, since the printing of the conductive inks on this film was not perfect, as previously mentioned, and when the test was carried out, the inks came off the substrate.

For carrying out the test on the traction equipment, specific test samples were designed and printed with the dimensions shown in Figure 10. The equipment was programmed to stretch the samples up to 10 mm, in intervals of 1 mm and with stops of 25 s, to carry out the electrical-resistance measurements.

Figure A7, Figure A8, Figure A9, Figure A10, Figure A11 and Figure A12 in Appendix A show the resistance variation according to the elongation of each pattern of the screen-printed inks on the different substrates.

Figure 11 shows the percentage variation of the resistance value according to the percentage elongation for the 1.5 mm-wide pattern, since it was the one used for the sensor design. The most suitable combination to obtain a reduced variation was the Inkron IPC-603X ink with the DelStar EU94DS substrate, as can be observed in the graph.

### 3.3. Study of the Conductive Tracks’ Resistance

The conductive tracks’ resistance after printing was 39.1 ± 0.2 Ω for Tu, 34.3 ± 0.5 Ω for Tp, 23.2 ± 0.4 Ω for Tr, and 21.0 ± 0.5 Ω for Ta. With these track resistance values; the substrate would have to be 133% longer to cause a variation of 0.5 °C in the measurement for the selected NTC thermistor.

### 3.4. Study of NTC Soldering

NTC solder with Inkron IPC-603X ink on a DelStar EU94DS substrate has been studied. The weld must not exceed 150 °C due to the PU melting point. This temperature limits the type of solder pastes that can be used, so work has been done with:The same silver ink (Inkron IPC-603X) that was used in the conductive tracks acting as a solder paste, although in this case it would act as a conductive adhesive. Curing was carried out at 130 °C. The advantage of soldering with conductive ink is the ability to use the same temperature in the conductor-manufacturing process, and in soldering, the disadvantage is that the ink viscosity is low, and it is difficult to place on pads with stencils.An anisotropic conductive adhesive that allows a first curing by long-wave ultraviolet rays (UVAs) to block the solder, and subsequently perform a low-temperature thermal curing. The paste used was DELO MONOPOX AC245 from DELO Adhesives (Windach, Germany). This paste requires a first UVA curing, between 320 nm and 400 nm from 1 s to 5 s, and then a thermal curing between 80 °C and 150 °C for 30 min and 10 min, respectively. In this case, a Ncure-Lab/Static 120 UV oven from EneMaq (Barcelona, Spain) with 0.5 J/cm^2^ and a FED-115 air oven from BINDER were used at 130 °C for 15 min. The advantages of the anisotropic solder paste are the proper viscosity and that the paste can be placed with less precision on the pads, while the disadvantage is that it requires two curing processes, UV and thermal.An NC-SMQ80/1E paste (52In/48Sn) from Indium Corp. (Clinton, NY, USA) with a melting point of 118 °C. The temperature peak was set at 135 °C for 110 s. This is a solder paste with a very low melting point, but with the drawback of not presenting Ag in its composition, making it difficult to solder with the silver ink.A Voltera T4/T5 (57.6Bi/42Sn/0.4Ag) from Voltera (Kitchener, Canada) with a melting point of 138 °C. The temperature peak was set at 145 °C for 90 s. This solder paste contains silver, and although its melting point is low, it was too close to the 150 °C limit imposed.

Two types of footprints were designed, one with the recommended size for solder reflow and the other for solder wave based on an encapsulation size of 0805 (Figure 12). All pastes and adhesives were printed on the track pads using a 250 μm-thick stainless-steel stencil with a print speed of 20 mm/s and a squeegee pressure of 0.02 kg/mm of blade length. Figure 13 and Figure 14 show the soldering results.

After printing and curing of the pastes, continuity measurements were made to check the correct soldering. Table 6 shows the percentages of correct solders.

With respect to the Inkron IPC-603X ink, the screen-printing technique was also used, since this ink is specific to this technology (Figure 15). In this case, percentages reached 100% and 85.7% for the solder reflow footprint and the wave solder footprint, respectively.

Figure 16 shows the appearance of the solder from the back of the PU. It was observed how the NC-SMQ80/1E paste had a very good integration with the conductor silver.

Once welded and protected with PDMS, the electrodes with the NTC thermistors must withstand a minimum pressure of 10 mmHg (1333.22 Pa) and a maximum of 40 mmHg (5332.9 Pa) by the compression bandage [21]. This means that in the 0805 package, a force of 0.015 N was on the Z axis, and a force of 0.006 N was on the X axis.

Table 7 shows the shear-strength tests performed on the Z, X, and X (rotation) axes for both types of footprint. Figure 17 shows the effects of the shear force on the solder reflow footprint on the X axis and X axis (rotation). It was observed that the break occurred simultaneously with the silver ink of the conductive tracks in the case of the NC-SMQ80/1E paste. This effect also was detected in the Voltera T4/T5 ink, although minor in the case of the X axis.

All results exceeded the needed values to withstand the pressure of the bandage, which was 0.015 N on the Z axis and 0.006 N on the X axis. However, the NC-SMQ80/1E paste significantly outperformed the rest in either case.

### 3.5. Field Testing

First, tests carried out in a healthy individual made it possible to verify the operation of the registration system. Once longitudinally separated (Figure A6, top), sensors were placed on healthy skin, ensuring good contact with the PDMS face. The NTC1 thermistor, the one with the longest connection, was located 3 cm from the ankle, leaving the others at different distances according to the length of the leg (Figure 18). Then, a minimally compressive bandage was applied, leaving the connection to the registration prototype out (Figure 19a). Finally, the system was fixed below the knee with a new padded bandage to protect the circuit and the leg (Figure 19b).

Figure 20 shows the evolution of the four temperatures involved over a 24 h period. The temperatures of the sensors in contact with the skin underwent small variations throughout the day, being very stable in the night period (hours 14 to 22 in the series). The differences between them remained in a very small range.

Figure 20 shows the temperature behavior in each of the regions where a sensor was placed in a simulated wound on the intact skin of an investigator. As mentioned above, the fluctuations between the sensors were minimal. These differences may vary, depending on, among other things, the activity that was being carried out at that time, the position of the leg, or the type of clothing. Between 10:00 p.m. and 3:00 p.m., the activity was minimal. From 3:00 p.m. to 10:00 p.m., the researcher underwent more activity as part of the test. It was found that the temperature increase occurred in all the sensors, especially in the NTC4 (environmental). The temperature change of the NTC1 between 5:00 a.m. and 8:00 a.m. could be caused by the position and support of the limb during the last hours of sleep. On the other hand, the increase of this same NTC thermistor in the most active time slot may have been due to the friction of the sensor with the bandage produced by the movement. Therefore, taking the activity into account is essential during the data collection to determine the cause of possible anomalous data.

## 4. Conclusions

In the development phase of the NTC electrode with encapsulation 0805, it was proven that, from a technological point of view, the best option was the use of a DelStar EU94DS medical grade polyurethane substrate with Inkron IPC-603X silver ink in screen printing; using the NC-SMQ80/1E paste from Indium for welding.

The NTC device must be protected with a PDMS layer. This structure can withstand the stresses of the compression bandage, as well as the inherent stretching in the use of the electrodes during the data-acquisition phase in a patient, without breaks.

The developed electronic system has proven to be robust, low consumption, and comfortable for the patient, and it allows control of the data acquisition times, as well as the data dump for further study.

Field tests in a healthy individual verified the correct operation of the system, as well as the data acquisition over long periods of time without causing discomfort to the individual and without electrode breaks or data losses.

The study’s next steps will consist of carrying out the necessary tests for cytotoxicity for contact with wounds, antimicrobials, measurements in patients, and the subsequent data treatment that will allow better monitoring of chronic wounds.

## Figures and Tables

**Figure 1 sensors-21-02741-f001:**
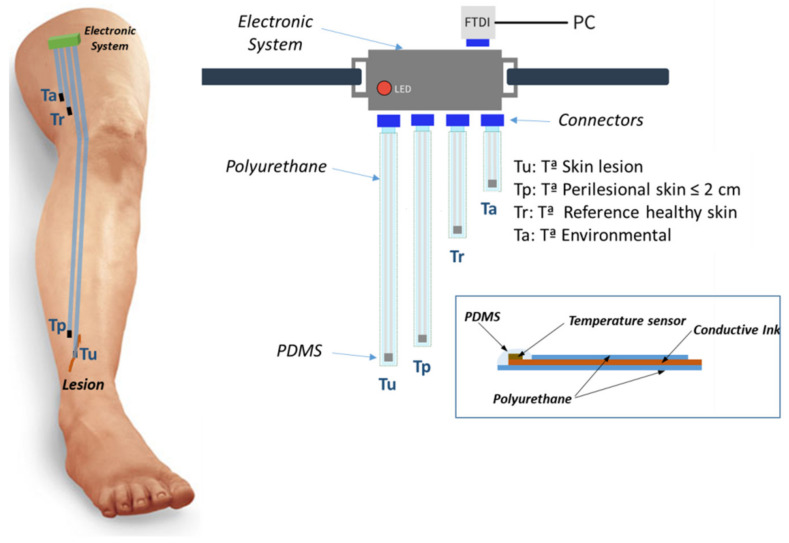
Sensor system configuration: the electronic measurement, data storage, and transmission system are attached to the leg by means of an extensible tape. The different temperature sensors placed in the areas detailed in the figure are connected to this system. The entire sensor system is covered with a compression bandage (not shown in the figure).

**Figure 2 sensors-21-02741-f002:**
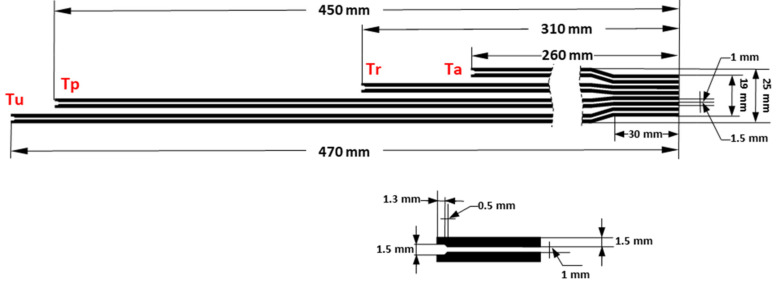
Multisensor dimensions. The top of the figure shows the overall dimensions, and the dimensions of the temperature sensor zone are shown at the bottom.

**Figure 3 sensors-21-02741-f003:**
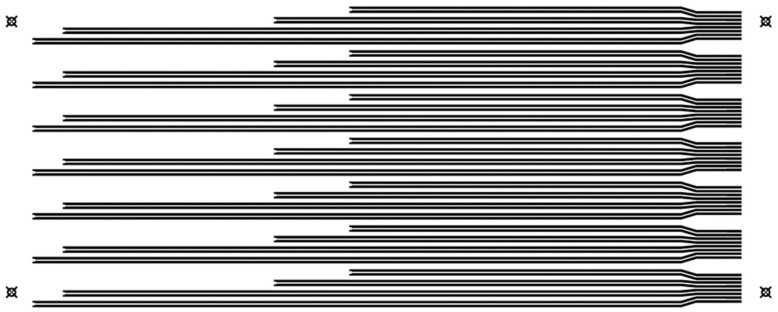
Pattern transferred to the printing screen. An 80 × 65 cm frame was used, so seven identical multi-sensors were included.

**Figure 4 sensors-21-02741-f004:**
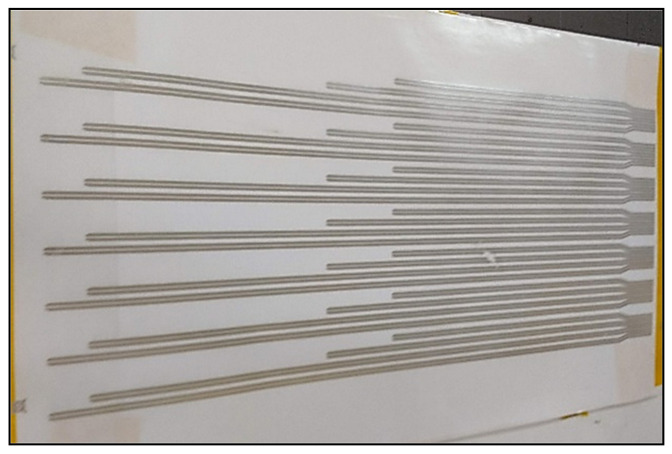
Pattern screen-printed with conductive silver ink on the polyurethane substrate.

**Figure 5 sensors-21-02741-f005:**
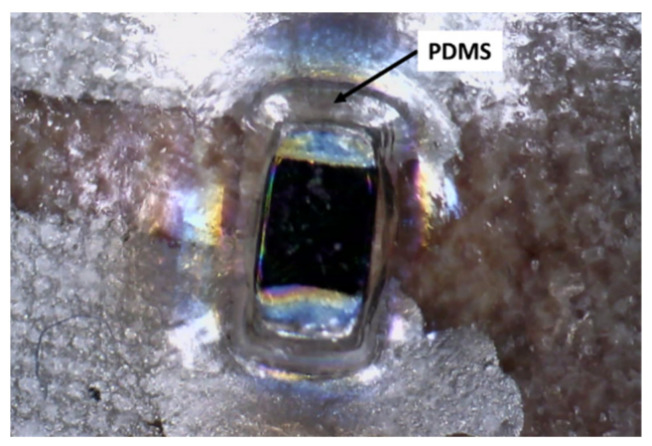
PDMS drop used to isolate the sensor from skin contact.

**Figure 6 sensors-21-02741-f006:**
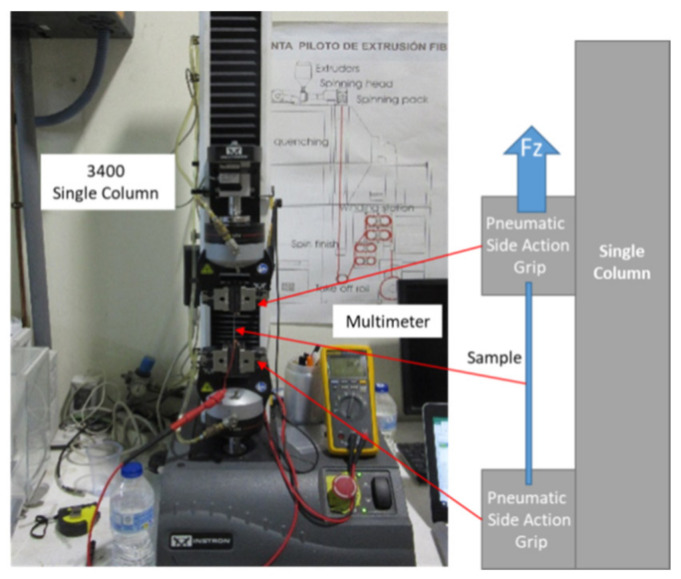
Experimental setup used for applying axial force and acquiring data during the electrode characterization.

**Figure 7 sensors-21-02741-f007:**
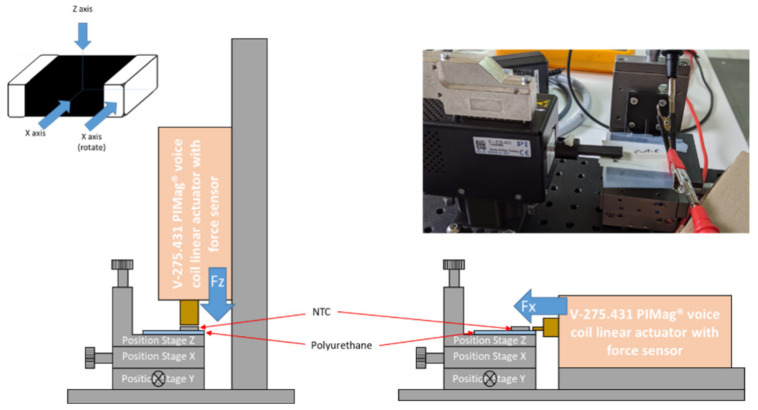
Experimental setup used for applying the Z and X forces (and X force rotation) and acquiring data during the electrode characterization.

**Figure 8 sensors-21-02741-f008:**
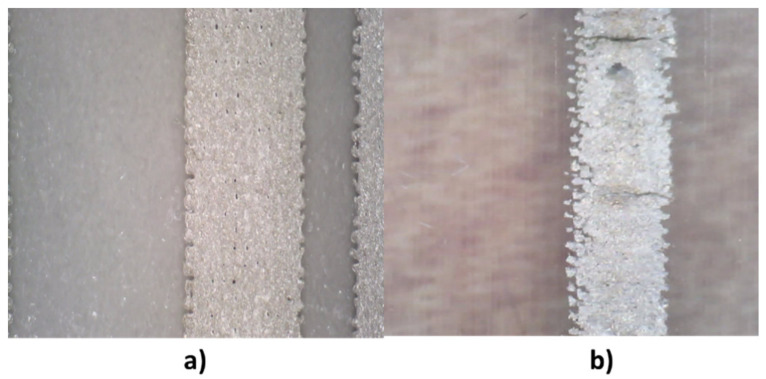
(**a**) Printing of the conductive silver ink on polyurethane; (**b**) printing defects of a conductive line on PDMS.

**Figure 9 sensors-21-02741-f009:**
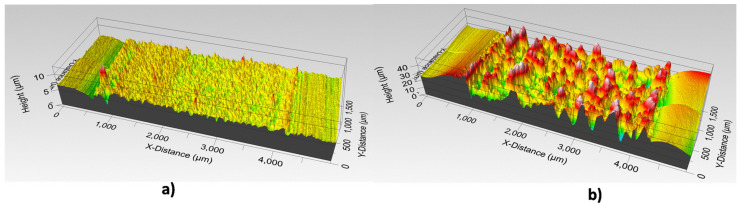
3D profilometry of a section of the pattern printed with (**a**) DuPont PE873 ink and (**b**) Inkron IPC-603X on the PET thermosetting substrate.

**Figure 10 sensors-21-02741-f010:**
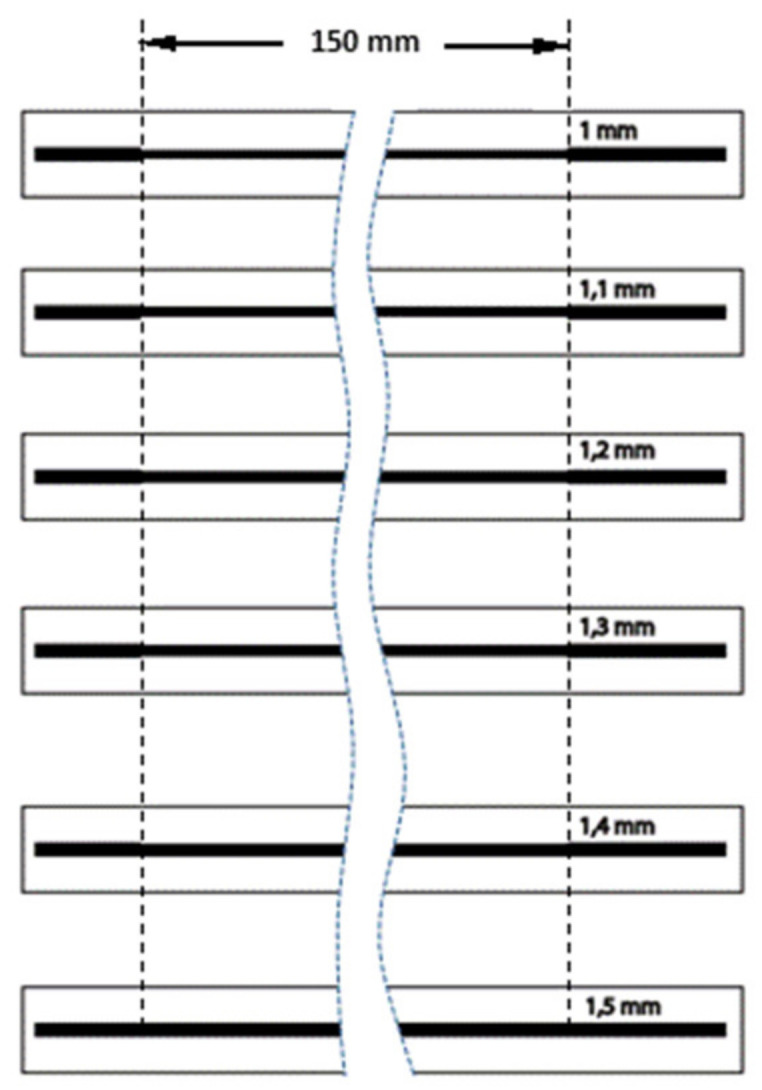
Dimensions of the pattern used to determine the parameters of elongation of inks and base films.

**Figure 11 sensors-21-02741-f011:**
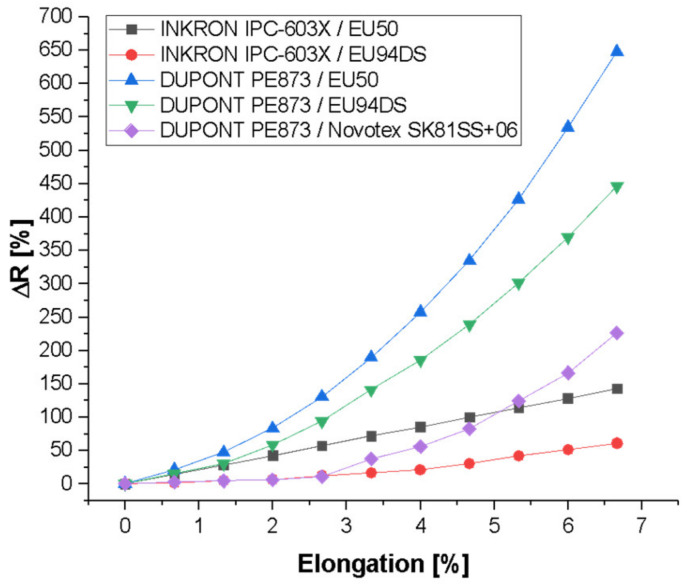
Percentage variation of the resistance value according to the percentage elongation for the 1.5 mm-wide pattern.

**Figure 12 sensors-21-02741-f012:**
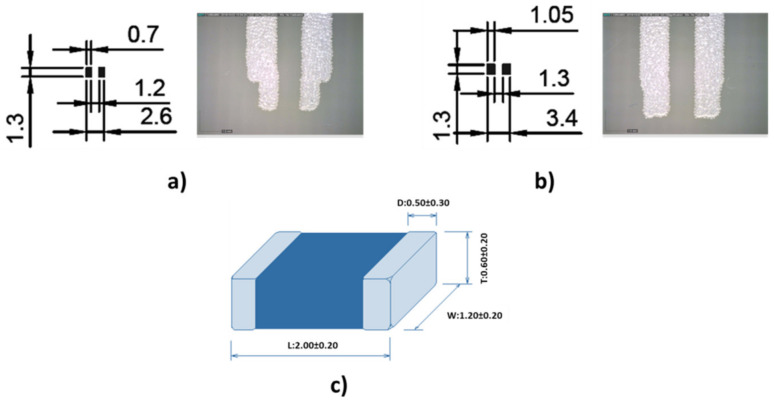
(**a**) The 0805 solder reflow footprint, (**b**) the 0805 wave solder footprint and (**c**) the 0805 package (mm).

**Figure 13 sensors-21-02741-f013:**
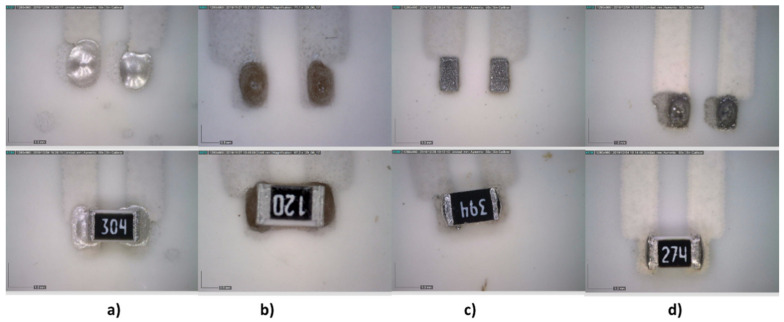
Stencil printing of pastes/adhesives on solder reflow footprint. Above: after printing; below: after temperature curing. (**a**) Inkron IPC-603X, (**b**) DELO MONOPOX AC245, (**c**) NC-SMQ80/1E, and (**d**) Voltera T4/T5.

**Figure 14 sensors-21-02741-f014:**
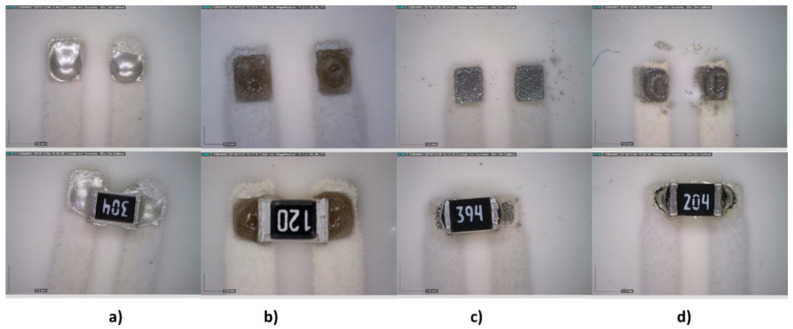
Stencil printing of pastes/adhesives on wave reflow footprint. Above: after printing; below: after temperature curing. (**a**) Inkron IPC-603X, (**b**) DELO MONOPOX AC245, (**c**) NC-SMQ80/1E, and (**d**) Voltera T4/T5.

**Figure 15 sensors-21-02741-f015:**
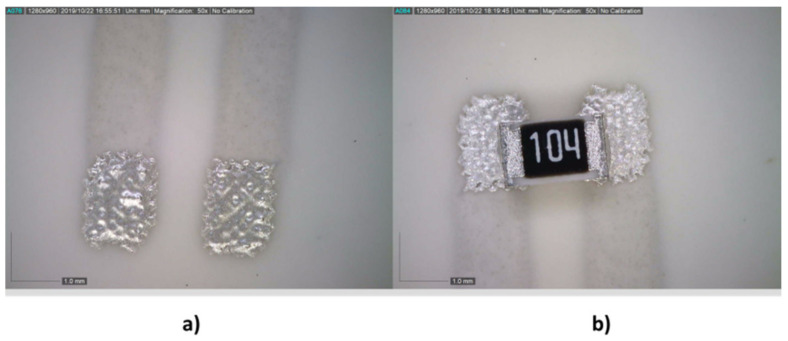
Screen printing of the Inkron IPC-603X on wave reflow footprint: (**a**) after printing; and (**b**) after temperature curing.

**Figure 16 sensors-21-02741-f016:**
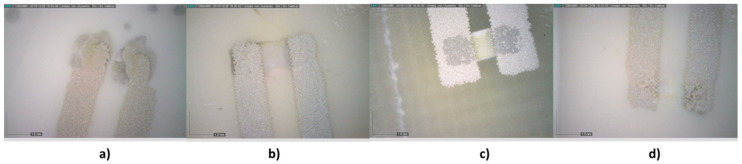
Solder view from the back of the polyurethane. (**a**) Inkron IPC-603X, (**b**) DELO MONOPOX AC245, (**c**) NC-SMQ80/1E, and (**d**) VOLTERA T4/T5.

**Figure 17 sensors-21-02741-f017:**
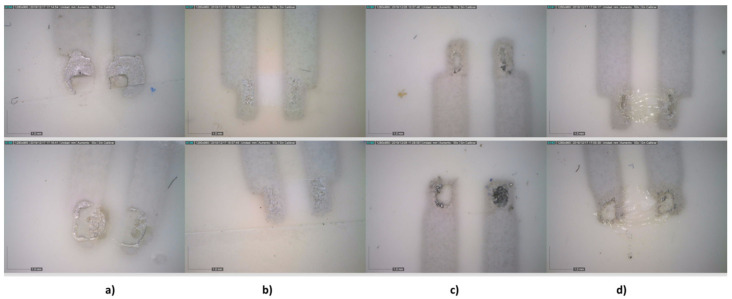
Effect of shear force on the solder reflow footprint: X axis (above) and X axis (rotation) (below). (**a**) Inkron IPC-603X, (**b**) DELO MONOPOX AC245, (**c**) NC-SMQ80/1E, and (**d**) Voltera T4/T5.

**Figure 18 sensors-21-02741-f018:**
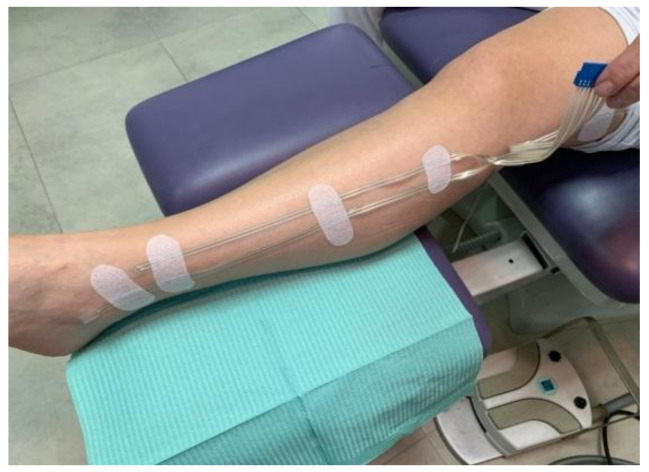
Placement of the system along the leg.

**Figure 19 sensors-21-02741-f019:**
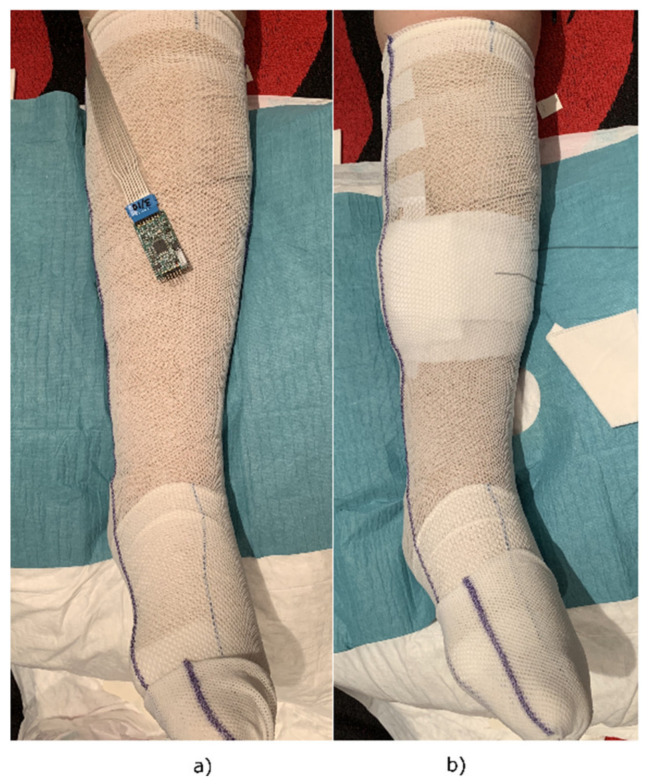
Placement of the system in the wound dressing. (**a**) Sensors applied under a finished bandage; (**b**) protection of the data acquisition system to avoid damaging the device and the patient during their activity.

**Figure 20 sensors-21-02741-f020:**
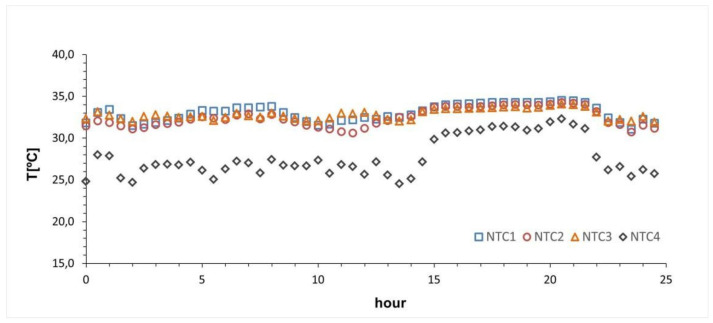
Temperature data recorded over a 24 h period. NTC1 = skin lesion, NTC2 = perilesional skin, NTC3 = healthy skin, and NTC4 = environmental.

**Table 1 sensors-21-02741-t001:** Characteristics of the prototype.

General	✓ Lightweight, small dimensions, with the ability to store information for later analysis ✓ Temperature sensor isolated by biocompatible systems allowing contact with healthy skin and the wound itself
Sensor Block	✓ Minimal impact on the cellular processes of the skin ✓ Adaptable to the anatomical morphology of the wound ✓ Reusable by sterilization ✓ Biocompatible materials to minimize the risk of allergic reactions and rejections ✓ Simple connectivity to the acquisition system ✓ Durability when it is in contact with the wound ✓ Individual sensors per measurement point
Electronic Block	✓ Small dimensions ✓ Lightweight ✓ Large data storage capacity ✓ Low energy consumption ✓ Data upload to PC via USB ✓ Simple software interface ✓ Real-time measurements

**Table 2 sensors-21-02741-t002:** Temperature sensor requirements.

Parameters	Specifications
Temperature resolution	0.1 °C
Support	Flexible and able to be adapted to the contour of the limb and the injury
Size	10 mm^2^ maximum
Encapsulation	Surface-mount technology (SMT)
Injury sensor location	In contact with the wound bed
Control sensor location	In contact with healthy/perilesional skin
Environmental sensor location	In the bandage, without skin contact
Power	Low energy
Material	Biocompatible

**Table 3 sensors-21-02741-t003:** Characteristics of the NTCS0805E3104SMT sensor.

Parameter	Value
Resistance value at 25 °C (Ω)	100 k
Tolerance on R25-value (%)	1
B25/85-value (K)	3590
Tolerance on B25/85-value (%)	±1
Maximum power dissipation (mW)	210
Response time (63.2%) 25 °C to 85 °C still air (s)	10
Dissipation factor δ in still air (mW/K)	3.5
Operating temperature range (°C)	−40 to +125
Weight (mg)	8
Dimensions L × W × T (mm)	2.0 × 1.25 × 0.8

**Table 4 sensors-21-02741-t004:** Characteristics of the films.

	DelStar EU50	DelStar EU94DS	Novotex SK81SS+06	PDMS
Thickness (µm)	50	80	25	100
Weight (g/m^3^)	55	94	1.20	1.12
MVTR * upright (g/m^2^/24 h)@37 °C	700	400	441	-
Tensile Strength MD ** (gf/cm)	1200	3000	3610	-
Elongation at break MD ** (%)	1000	700	511	570

* Moisture vapor transmission rate (MVTR). ** Machine direction (MD).

**Table 5 sensors-21-02741-t005:** Characteristics of the silver inks.

	DUPONT PE873	INKRON IPC-603X
Sheet resistivity (mΩ/sq/mil)	<75	<15
Solids (%)	60–65	100
Viscosity (Pas)	50–80 @0.2 s^−1^	16 @0.25 s^−1^
Screens polyester (threads/inch)	120–77	
Curing	160 °C—10 min	130 °C—15 min
Properties	Stretchable Flexible Washable with encapsulation	High stretchability Flexible

**Table 6 sensors-21-02741-t006:** Percentage of correct package soldered with stencil printing technique.

	Solder Reflow Footprint	Wave Solder Footprint
INKRON IPC-603X	57.1%	57.1%
DELO MONOPOX AC245	71.4%	28.5%
NC-SMQ80/1E	100%	100%
VOLTERA T4/T5	71.4%	57.1%

**Table 7 sensors-21-02741-t007:** Maximum force (N) applied on the axis.

	Z axis		X axis		X axis (Rotation)	
	Solder Reflow Footprint	Wave Solder Footprint	Solder Reflow Footprint	Wave Solder Footprint	Solder Reflow Footprint	Wave Solder Footprint
INKRON IPC-603X	10	10	0.304	0.809	0.433	1.247
DELO MONOPOX AC245	10	0.01	0.787	0.001	0.377	0.001
NC-SMQ80/1E	10	10	1.247	1.625	1.119	1.496
VOLTERA T4/T5	10	10	0.156	0.375	0.287	0.896

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
