# Peer review of "Flexible Hybrid Electrodes for Continuous Measurement of the Local Temperature in Long-Term Wounds"

_sensors, 2021, doi:10.3390/s21082741_

Round 1

Reviewer 1 Report

Overall a very interesting and thorough piece of work. I feel that this work will be of interest to the readership of Sensors. I recommend Accept after Minor Revisions (listed below).

1) The introduction would benefit from the addition of some references of the recent work on stretchable and wearable electronic diagnostic devices by Kassasnos et al.

  • Panagiotis Kassanos, Bruno Gil Rosa, Meysam Keshavarz, Guang-Zhong Yang, Chapter 2 - From wearables to implantables—clinical drive and technical challenges, Editor(s): Edward Sazonov, Wearable Sensors (Second Edition), Academic Press, 2021, Pgs. 29-84
  • P. Kassanos, F. Seichepine, D. Wales and G. Yang, "Towards a Flexible/Stretchable Multiparametric Sensing Device for Surgical and Wearable Applications," 2019 IEEE Biomedical Circuits and Systems Conference (BioCAS), Nara, Japan, 2019, pp. 1-4, doi: 10.1109/BIOCAS.2019.8919197.
  • P. Kassanos, F. Seichepine and G. -Z. Yang, "Characterization and Modeling of a Flexible Tetrapolar Bioimpedance Sensor and Measurements of Intestinal Tissues," 2019 IEEE 19th International Conference on Bioinformatics and Bioengineering (BIBE), Athens, Greece, 2019, pp. 686-690, doi: 10.1109/BIBE.2019.00129.

2) Have the authors considered a wireless interface for data communiction with the device?

3) For better understanding for the readers, the authors should define "perilesional" in the introduction.

4) Figure 8 - (a) and (b) need to be added to the figure to distinguish which image is which.

Author Response

The authors would like to thank the Reviewer for the deep and thorough review of this manuscript. The research paper has been revised in the light of the Reviewer’s useful suggestions and comments, hoping to have improved its quality to a better scientific level.

Please, see the attached document. 

Reviewer 2 Report

This paper proposed a portable prototype that allows to know the local temperature behaviour of a long term wound over time. The use of flexible substrates, screen-printing techniques with polymeric inks and embedded systems are allowing the development of biomarkers. I have some comments.

  1. The main goals of this paper is fabricated screen-printing silver line on flexible substrates and embeded the temperature measurement system. However, the abstract is not provide the main purposes of this manuscript. This will mislead readers to read this paper.
  2. In Appendix, the resistance was increased at large elongation. However, the length of injury temperature sensor is larger than others. Therefore, the higher temperature was obtained at injury temperature as excepted. The length select is important. Tow to select the length of four sensors?
  3. The authors should provide the resistance error bar of length of silver line.
  4. Fig. 20: Only the little higher temperature for NTC1 in 5-8 and 16-20 hr. some of the NTC1 is lower than NTC3. The temperature for four sensors should be discussed.
  5. I think the paper should be take care to rewrite. Although more results were shown. But the novelty and proposed are confused for readers. I suggest that the PDMS is only used for drop to isolate the SMD and skin, thereby, the Fig. 8 can be removed and some parameters of PDMS films also removed.
  6. The resistance of various length of silver line for temperature should be provided.

Author Response

(The authors gave the same response as above.)

Round 2

Reviewer 2 Report

The authors have properly answered all my comments and now it can be accepted.